# Spatiotemporal variation of malaria incidence in parasite clearance interventions and non-intervention areas in the Amhara Regional State, Ethiopia

**Melkamu Tiruneh Zeleke** [1] *, **Kassahun Alemu Gelaye**[2], **Muluken Azage Yenesew**[1]

**1** School of Public Health, College of Medicine and Health Sciences, Bahir Dar University, Bahir Dar, Ethiopia, **2** Institute of Public Health, University of Gondar, Gondar, Ethiopia

* mtiruneh089@gmail.com

**Data Availability Statement:** Due to third-party restrictions, data are available from the Amhara Public Health Institute (APHI). Interested researchers may contact the institute to get

## Abstract

### Background

In Ethiopia, malaria remains a major public health problem. To eliminate malaria, parasite clearance interventions were implemented in six kebeles (the lowest administrative unit) in the Amhara region. Understanding the spatiotemporal distribution of malaria is essential for targeting appropriate parasite clearance interventions to achieve the elimination goal. However, little is known about the spatiotemporal distribution of malaria incidence in the intervention and non-intervention areas. This study aimed to investigate the spatiotemporal distribution of community-based malaria in the intervention and non-intervention kebeles between 2013 and 2018 in the Amhara Regional State, Ethiopia.

### Methods

Malaria data from 212 kebeles in eight districts were downloaded from the *District Health Information System2* (*DHIS2*) database. We used Autoregressive integrated moving average (ARIMA) model to investigate seasonal variations; Anselin Local Moran's I statistical analysis to detect hotspot and cold spot clusters of malaria cases; and a discrete Poisson model using Kulldorff scan statistics to identify statistically significant clusters of malaria cases.

### Results

The result showed that the reduction in the trend of malaria incidence was higher in the intervention areas compared to the non-intervention areas during the study period with a slope of -0.044 (-0.064, -0.023) and -0.038 (-0.051, -0.024), respectively. However, the difference was not statistically significant. The Global Moran's I statistics detected the presence of malaria clusters (z-score = 12.05; p<0.001); the Anselin Local Moran's I statistics identified hotspot malaria clusters at 21 locations in Gendawuha and Metema districts. A statistically significant spatial, temporal, and space-time cluster of malaria cases were detected. Most likely type of spatial clusters of malaria cases (LLR = 195501.5; p <0.001) were detected in all kebeles of Gendawuha and Metema districts. The temporal scan statistic identified three peak periods

permission to access the data. You can contact the institute through email: admin@aphi.gov.et or aphi172008@gmail.com; Office phone number: +251582263227.

**Funding:** The author(s) received no specfic funding for this work.

**Competing interests:** The authors have declared that no competing interests exist.

**Abbreviations:** ACF, Autocorrelation Function; ARIMA, Autoregressive Integrated Moving Average; APHI, Amhara Public Health Institute; CMA, Centered Moving Average; CSA, Central Statistics Agency; DHIS2, District Health Information System 2; FTAT, Focal Testing and Treatment; GPS, Global Positioning System; IR, Incidence Rate; IRS, Indoor Residual Spraying; LLR, Log-Likelihood Ratio; LLINs, Long-Lasting Insecticide Treated Nets; MDA, Mass Drug Administration; MTAT, Mass Testing and Treatment; PACF, Partial Autocorrelation Function; PATH, Program for Appropriate Technology in Health; RDT, Rapid Diagnostic Test; RR, Relative Risk; WHO, World Health Organization.

between September 2013 and November 2015 (LLR = 8727.5; p<0.001). Statistically significant most-likely type of space-time clusters of malaria cases (LLR = 97494.3; p<0.001) were detected at 22 locations from June 2014 to November 2016 in Metema district.

## Conclusion

There was a significant decline in malaria incidence in the intervention areas. There were statistically significant spatiotemporal variations of malaria in the study areas. Applying appropriate parasite clearance interventions is highly recommended for the better achievement of the elimination goal. A more rigorous evaluation of the impact of parasite clearance interventions is recommended.

## Introduction

Despite the availability of effective vector control strategies, malaria remains a major public health problem in 85 malaria-endemic countries and territories including Ethiopia. Globally, an estimated 241 million cases of malaria were reported in 2020; an additional 14 million cases were reported as compared to 2019 [1–3]. In 2015, the World Health Organization (WHO) set new goals of reducing global malaria case incidence and mortality rate by 90% and eliminating malaria in 35 countries by 2030. Ethiopia has been launched its malaria elimination program in 2017 to achieve elimination within the same period [4–7].

Malaria control efforts have been focused on vector control strategies such as Long Lasting Insecticide Treated Nets (LLINs), Indoor Residual Spraying (IRS), and environmental management to reduce adult mosquito populations and human mosquito contact and eradicate mosquito breeding inhabitants [8]. However, to achieve malaria elimination, parasite clearance interventions are essential to clear both symptomatic and asymptomatic infections in the human population [8–10]. Malaria parasite clearance interventions with antimalarial drugs are potentially a useful tool to eliminate malaria. Mass drug administration (MDA), mass testing and treatment (MTAT), and focal testing and treatment (FTAT) are among the widely used parasite clearance interventions [11–17].

In a collaboration between the government and partners, mass testing and treatment followed by focal testing and treatment interventions were implemented in selected six kebeles with having different malaria transmission intensities in the Amhara Regional State, Ethiopia between August 2014, and September 2018. The coverage of the interventions was found above 80% and it is considered a feasible intervention in Ethiopia [12, 17]. Besides the parasite clearance interventions, a weekly kebele-based malaria report was collected at the intervention and non-intervention kebeles from Epi week 37 of 2013 through 38 of 2018 using the *DHIS2* platform.

The distribution of infectious diseases shows marked heterogeneity [18, 19]. This heterogeneity reduces the efficacy of disease control and elimination strategies [20]. The distribution of malaria and other infectious diseases has been investigated to understand the distribution dynamics using different spatial statistical tools [21–39]. Identifying a cluster of malaria cases is used for targeting appropriate malaria control and elimination interventions including parasite clearance interventions [18, 20, 21, 28, 35, 40]. Although rigorous studies have been conducted to investigate the spatiotemporal variations of malaria at the global, regional, and local scales, the studies didn't consider the malaria transmission intensities in their analysis.

In Ethiopia, little is known about the spatiotemporal distribution of malaria at the community level including in the parasite clearance interventions and non-intervention kebeles. Therefore, this study aimed to investigate the spatiotemporal distribution of community-based malaria using the data generated from *District Health Information System2* (*DHIS2*) in the intervention and non-intervention kebeles by considering the transmission settings in the Amhara Regional State, Ethiopia.

## Materials and methods

### Study areas

The study was conducted in 212 kebeles (the lowest administrative unit in Ethiopia) under eight districts in the Amhara Regional State, Ethiopia (Fig 1). The study districts are found in four different ecological-epidemiological settings. High malaria transmission settings (Gendawuha and Metema), moderate transmission settings (Bahir Dar Zuria and Mecha), low transmission settings (Kalu and Tehulederie), and very low (Aneded and Awabel) [41]. According to the Central Statistics Agency (CSA), Ethiopia, an estimated 1.4 million population reside in the study area [42]. Malaria transmission in the study districts is seasonal and unstable, the major malaria transmission season from September through December following the major rainy season from June to August. The minor transmission season is from April to June following the minor rainy season, February and March [43]. The recorded daily temperature of the study areas showed an average minimum temperature of 13.3˚c, and an average maximum temperature of 31.3˚c [44].

Parasite clearance interventions with antimalarial drugs were implemented in six kebeles having different malaria transmission intensities. The intervention kebeles were Dehina Sositu

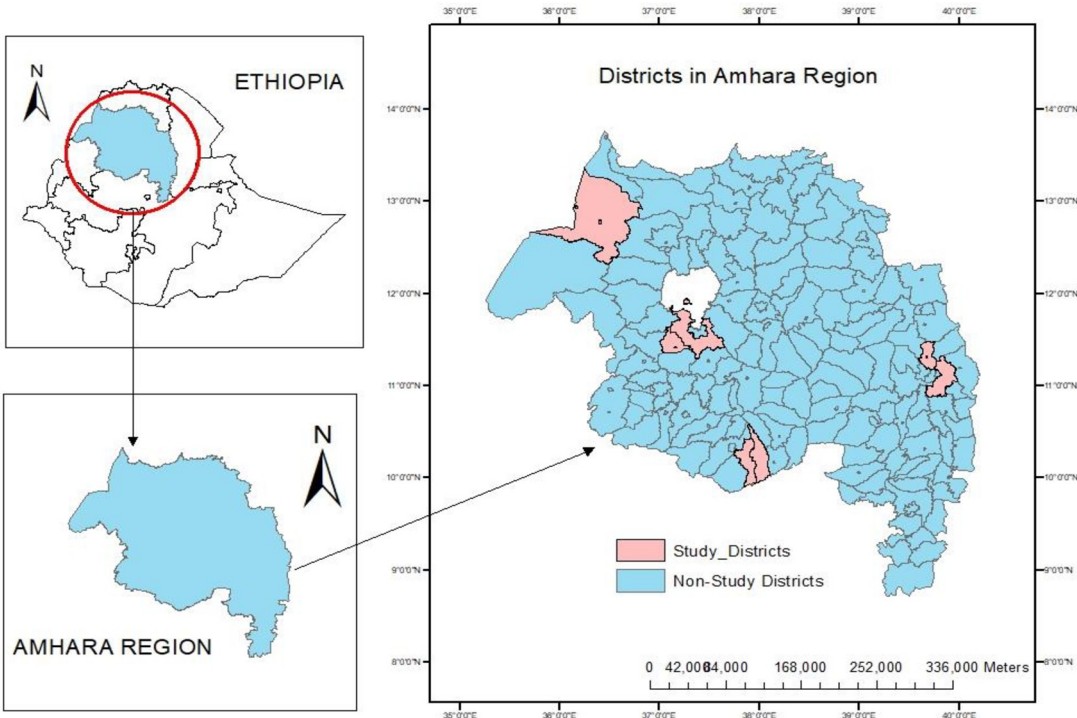

**Fig 1. Map of study districts in the Amhara Regional State, Ethiopia. Copyright:** © 2022 Zeleke et al. The data on the map is derived from CSA, Ethiopia and APHI, Bahir Dar.

and Yeginid Lomi in Bahir Dar Zuria District, Berhan Chora in Mecha District, Zengoba in Aneded District, Choresa in Kalu District, and Kumer Aftit in Metema District. All other kebeles in each district were non-intervention [12].

## Data

Malaria data were downloaded from the *DHIS2* database between epi week 37/2013 and 38/2018. Disaggregated malaria data by kebele were collected using a smart mobile phone from the health posts and health centers disaggregated by kebeles. The data elements include total outpatients, total patients suspected and tested for malaria either microscopy or rapid diagnostic test (RDT), total *Plasmodium Falciparum*, *Plasmodium Vivax*, and Mixed. The data quality was monitored every week through the validation rule set in the *DHIS2* platform. The geographic coordinates (Altitude, Latitude, and longitude) of each kebele were collected using a hand-held global positioning system (GPS) with an accuracy of less than 5. The mid-year projected population of each kebele and the shapefiles were obtained from the CSA, Ethiopia.

## Data analysis

**Trend and seasonal analyses.**   The weekly, monthly, and annual malaria incidence of each kebeles were calculated and plotted to check the variation of malaria transmission between September 2013 and September 2018. The number of malaria cases reported to the population at risk was used to calculate the malaria incidence in the specified period.

Seasonal decomposition analysis was conducted using the autoregressive integrated moving average (ARIMA) model to evaluate the seasonal variation, irregularity, and trend components of time-series malaria data in the study areas. Autocorrelation function (ACF) and partial autocorrelation function (PACF) charts were used to determine which model and order to be used. Smoothing was performed to remove any seasonal and short-term variations from the dataset for suitable trend analysis. A multiplicative model was used for the analysis which is the product of time-series components [45].

**Spatial cluster analysis.**   Two methods of spatial clustering analyses were employed to detect clusters of malaria. The first method was Global Moran's I statistic (spatial autocorrelation) using ArcGIS 10.8. This method was employed to examine the presence of spatial autocorrelation across the entire dataset. The critical distance was determined using the incremental Global Moran's I. The Global Moran's I statistical analysis tests the null hypothesis that measures the values at a location independent of values with other locations, the values vary from -1 to 1. Positive (negative) values indicate the presence of positive (negative) spatial autocorrelation, whereas a zero value indicates a random spatial pattern [46].

Anselin Local Moran's I statistic was used to detect/map hotspot and cold spot clusters and outliers. Hotspot spatial clusters of malaria were identified by detecting local areas where high incidence kebeles border with other high incidence kebeles (high-high) and cold spot spatial clusters of malaria were identified by detecting local areas where low incidence kebeles border with other low incidence kebeles (low-low). Outliers were identified by detecting local areas where high incidence kebeles border with low incidence kebeles and vice versa [47, 48].

The second method was Kulldorff's spatial scan statistics using SaTScan^TM version 10.0 software. Kulldorff scan statistics method was used to identify statistically significant spatial, temporal, and space-time clusters of malaria cases. A discrete Poisson model was used as the number of malaria cases in each location was a count data and Poisson distributed. Patients with malaria were taken as cases, and the mid-year population was taken as at risk of malaria. Then, a discrete Poisson model was run to analyze the purely spatial, temporal, and space-time scan statistics [49].

The scan statistics were used to detect a cluster of cases i.e., areas with a larger number of cases than would be expected by chance. This indicates areas where there may be a higher risk of malaria. SaTScan™ imposes circular windows of varying sizes on the spatial data to detect statistically significant clusters of malaria cases. In this study, the maximum spatial cluster size of the population at risk was set to 25% to 50% depending on the ecological-epidemiological settings. The observed cases were compared with the expected cases inside and outside each window, and the risk ratios were estimated based on Poisson distribution.

A statistically significant cluster was investigated with a log-likelihood ratio test using the number of Monte Carlo replication which was set to 999 (the default) since the dataset was relatively large. The minimum number of cases was restricted from two to five and the relative risks were restricted from 1 to 1.5 to identify clusters of malaria cases by considering the malaria transmission intensity. The method was used to identify not only the most likely significant clusters but also significant secondary clusters [50]. For purely spatial and space-time analyses, in addition to the most likely clusters, secondary clusters were identified and ordered according to their log-likelihood ratio result.

**Temporal and spatiotemporal cluster analysis.** The reported weekly malaria cases were aggregated into monthly to analyze the purely temporal and space-time clusters of malaria cases since SaTScan™ version 10.0 software lacks the weekly time precision. Retrospective purely temporal cluster analysis with high rates using the discrete Poisson model was used to detect the temporal clusters of malaria cases. In this study, the time aggregation unit was a month (with a length of one month) and the maximum temporal window size was set to 50% of the study period as a temporal cluster. The maximum number of Monte Carlo replications was set to 9999. For the purely temporal analysis, only the most likely cluster was reported.

The space-time cluster was detected with high rates through the retrospective space-time scanning using the discrete Poisson model. The maximum temporal cluster size for space-time scan statistics was used 50% for the whole study period. The space-time scan statistics were defined by a cylindrical window with a circular geographic base and with height corresponding to time. The maximum number of Standard Monte Carlo replication was set to 9999. Space-time cluster analysis was used to identify both the most likely and secondary significant clusters of malaria cases [51].

The procedure for the purely spatial and space-time cluster analyses was set to report the most likely cluster/s in the first iteration and then the most likely cluster/s removed from the dataset. In the second iteration, the first statistically significant secondary cluster/s is/are reported and removed from the remaining dataset. This procedure was then repeated until there was no more cluster with a p-value less than 0.05. Statistical analyses were performed using ArcGIS Version 10.8, SaTScan™ version 10.0, IBM SPSS version 23, and MS Excel software.

## Ethical considerations

Ethical clearance was obtained from the Institutional Review Board (IRB) of the College of Medicine and Health Sciences, Bahir Dar University with protocol number 00223/2020. A letter of support from Bahir Dar University was written to the Amhara Public Health Institute (APHI) to access and use the retrospective data. The data were collected and aggregated by kebele level, no individual identifiers were attached to the data, and all the information was kept confidential.

## Results

### Trend and seasonal analyses of malaria

Two hundred twelve kebeles in eight districts were included in this study. Between epi week 37/2013 and 38/2018, a total of 175,350 malaria cases were reported from the study areas.

*Plasmodium falciparum* (66.6%) species was the dominant followed by *Plasmodium vivax* (25.4%) and mixed infections (8%). The highest incidence of malaria (39.5 per 1000 population at risk per week) occurred in Meka kebele, Metema district during epi week 38 of 2016.

Malaria incidence during the study period showed a declining trend both in the intervention and non-intervention kebeles. Seasonal variation of malaria transmission was observed. In November 2015, the seasonality and irregularity components were 80% above the baseline (centered moving average, CMA) in the intervention kebeles. The peak malaria incidence was observed from October to December 2013 and October 2014 in the intervention kebeles (Fig 2).

In October 2016, the seasonality and irregularity components were 54% above the baseline in the non-intervention kebeles. Multiple peaks of malaria incidence were observed during October throughout the study period in the non-intervention kebeles (Fig 3).

## Spatial cluster analysis

**Spatial autocorrelation analysis.** Global Moran's I statistic detected the presence of malaria clusters with a z-score of 12.05 and p-value <0.001; there is a less than 1% likelihood that this clustered pattern could be the result of random chance (Fig 4).

**Hotspot/cold spot analysis.** Anselin Local Moran's I statistic identified 21 hotspot clusters of malaria cases in Gendawuha and Metema districts. The locations were Awasa, Awlala, Das Michael, Diviko, Gendawuha Town 01, Gendawuha Birshign, Gubay Jejebit, Kokit Town, Kumer Aftit, Lemlem Terara, Lencha, Meka, Mender 6 7 8, Metemayohannes 01, Metemayohannes 02, Metemayohannes 03, Shemlegara, Tagur, Tumet, Wodi Anbeso, Zebach Bahir (Fig 5).

**Purely spatial clusters of malaria cases.** In the study areas, malaria cases were not randomly distributed. The purely spatial cluster analysis identified one most likely type of cluster with 28 locations and six secondary significant clusters with eight locations. The most likely type of cluster of malaria cases (log-likelihood ratio (LLR) = 195501.5; p-value <0.001) was detected in the Gendawuha and Metema districts. The cluster window was centered at

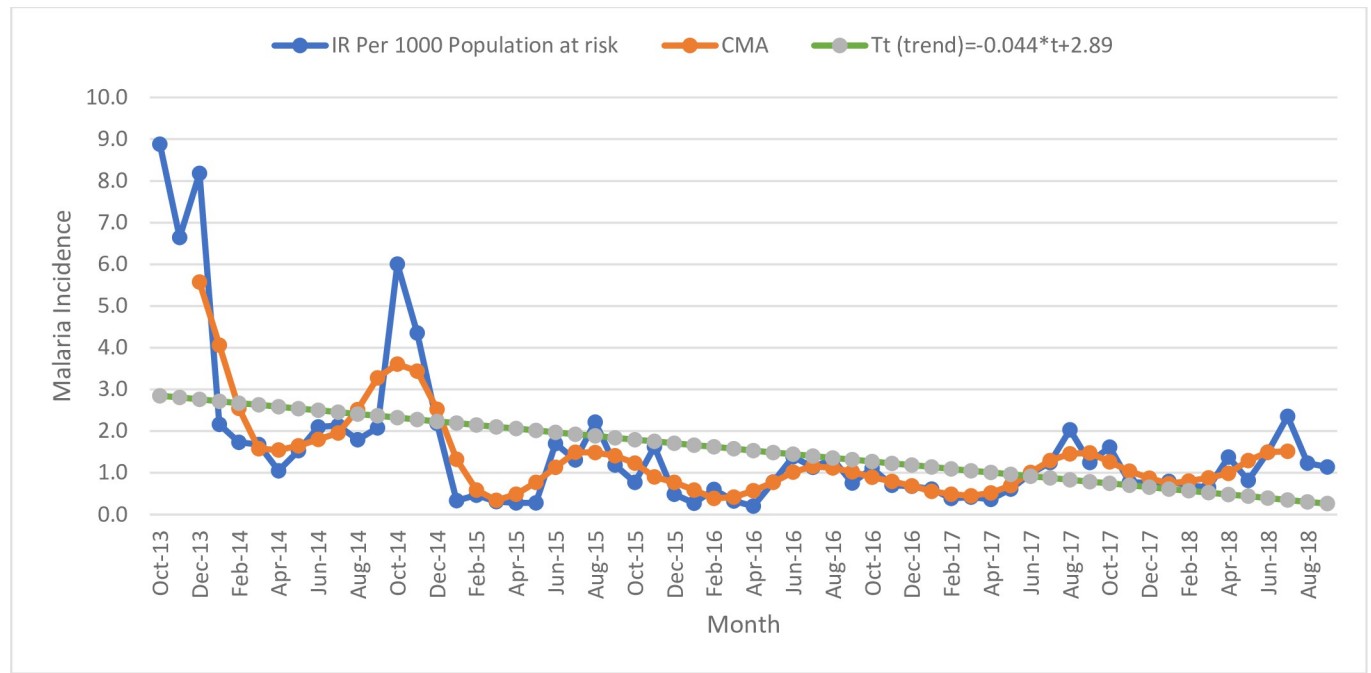

**Fig 2. Seasonal decomposition of malaria incidence per 1000 population at risk between September 2013 and 2018 in the intervention kebeles.**

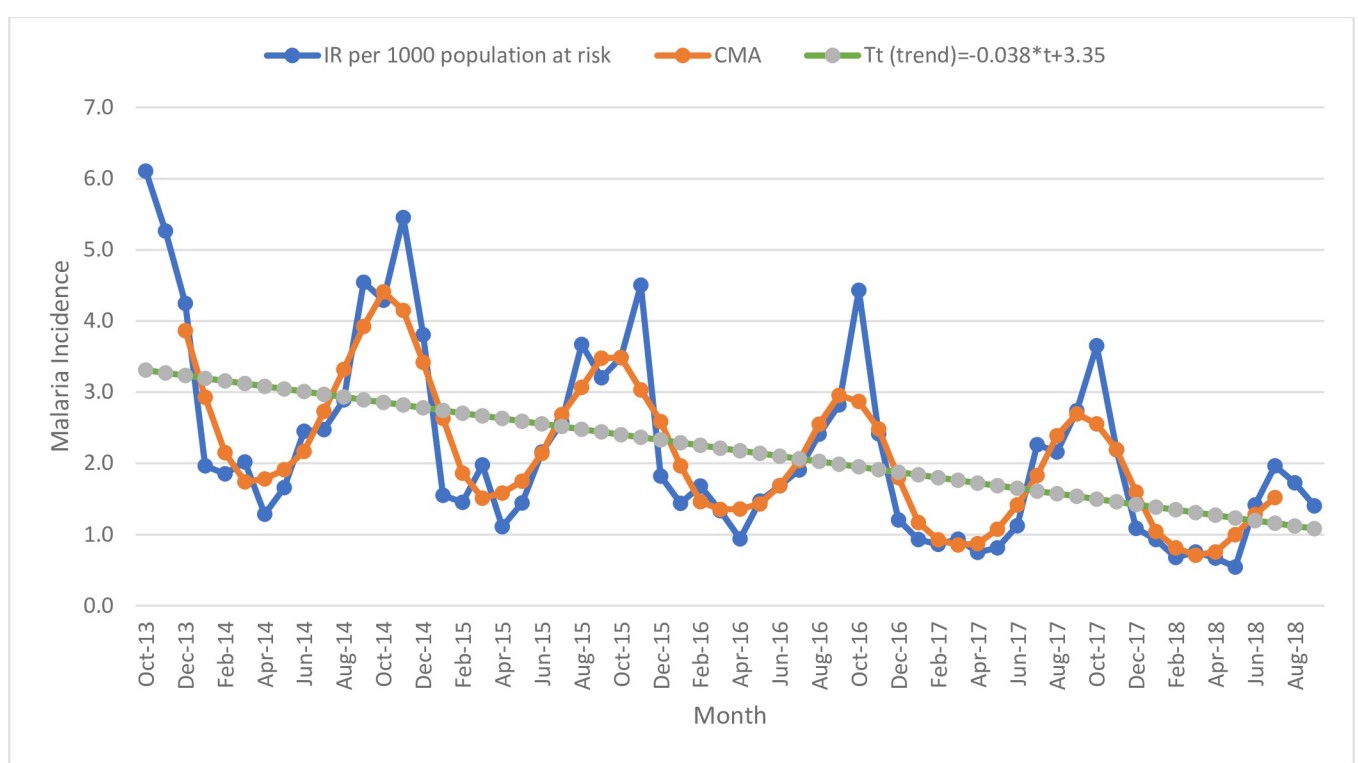

**Fig 3. Seasonal decomposition of malaria incidence per 1000 population at risk between September 2013 and 2018 in the non-intervention kebeles.**

12.863793N, 36.716339E (Achera kebele) with 28 locations around an 81.7 km radius. Significant secondary clusters of malaria cases were detected in Aneded, Awabel, Bahir Dar Zuria, Kalu, and Mecha districts (Table 1 and Fig 6).

Many more most likely and secondary significant clusters of malaria cases were detected when a separate analysis was done by considering the different ecological-epidemiological settings in the study areas. The separate spatial cluster analysis of Gendawuha and Metema

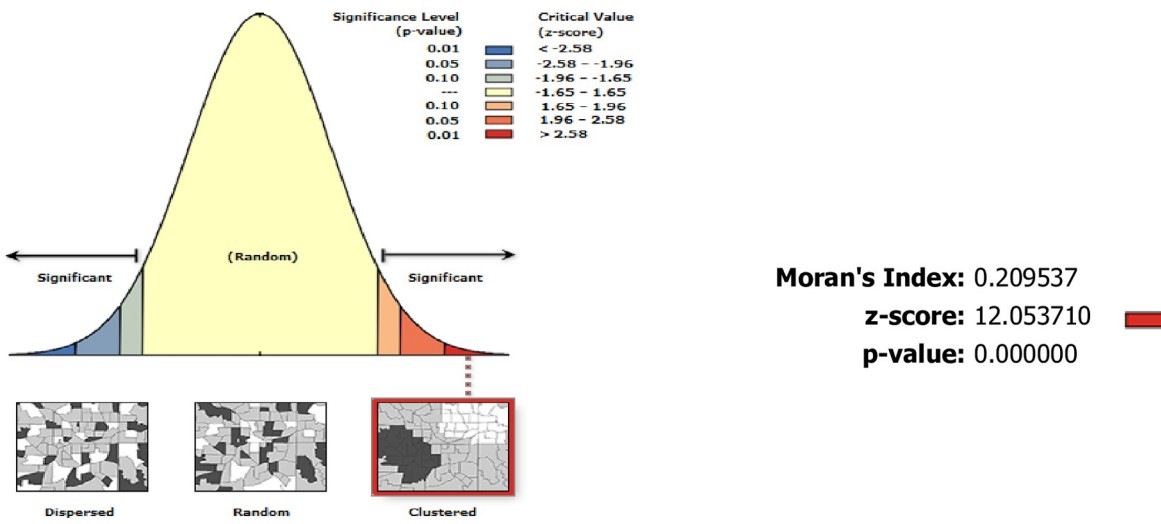

**Fig 4. Global Moran's I spatial autocorrelation report.**

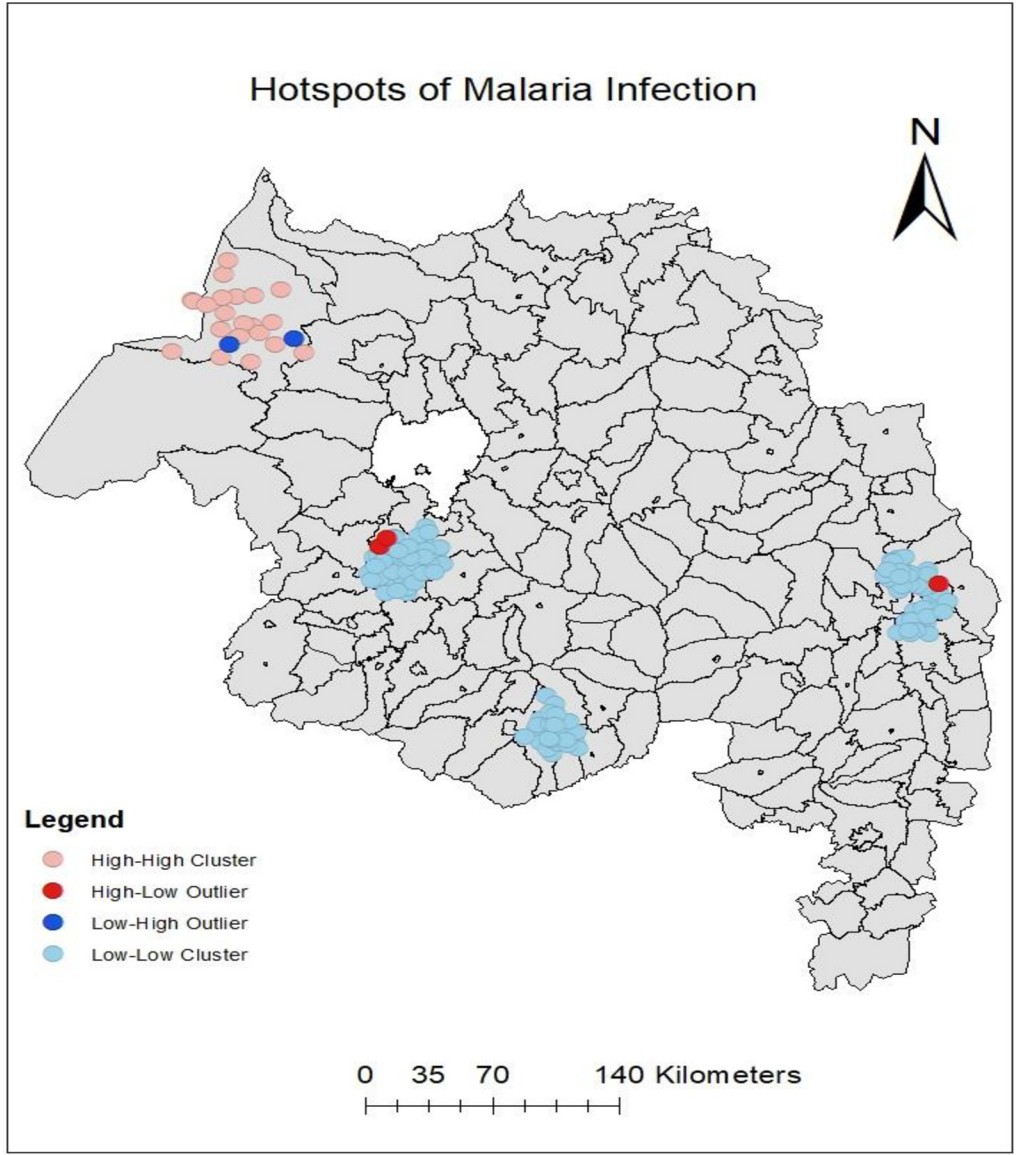

**Fig 5. Hotspot clusters of malaria cases in Metema and Gendawuha districts during the study period. Copyright:** © 2022 Zeleke et al. The data on the map is derived from CSA, Ethiopia and APHI, Bahir Dar.

districts identified one most likely type of cluster (LLR = 12541.5; p-value <0.001) with a single location (Meka Kebele) and two secondary significant clusters with six locations at Metema district (Table 2).

The spatial cluster analysis of Bahir Dar Zuria and Mecha districts identified one most likely type of cluster of malaria cases (LLR = 1599.1; p-value <0.001) was detected with four locations centered at Yeginid kebele (11.443215N, 37.565428E/10.30 km). Thirteen secondary significant clusters with 21 locations were identified in the two districts (Table 3).

In Kalu and Tehulederie districts, the spatial cluster analysis identified one most likely type of cluster (LLR = 1380.1; p-value <0.001) with a single location at Jerjero (023) kebele (11.209294N, 39.904368E/ 0 km). Eleven secondary significant clusters with 13 locations were identified in the two districts (Table 4).

**Table 1. Purely spatial clusters of malaria cases in the study areas between epi week 37/2013 and 38/2018.**

| Cluster type | District | Kebele | Coordinates/Radius | Locations | Obs. cases | Exp. cases | RR | LLR | P-value |
|---|---|---|---|---|---|---|---|---|---|
| Most likely cluster | Metema/ Gendawuha | Achera* | 12.863793N,36.716339E/81.7km | 28 | 137553 | 22624.2 | 24.57 | 195501.5 | <0.001 |
| Secondary cluster1 | Awabel | Dimamelese** | 10.049228N, 38.016623E/4.9km | 3 | 3853 | 1982.8 | 1.96 | 699.7 | <0.001 |
| Secondary cluster2 | Aneded | Malgash | 10.031071N, 37.918553E/0km | 1 | 1985 | 917.3 | 2.18 | 467.9 | <0.001 |
| Secondary cluster3 | Aneded | Shumburma | 10.065235N, 37.897291E/0km | 1 | 2139 | 1264.9 | 1.70 | 251.8 | <0.001 |
| Secondary cluster4 | Mecha | Tekle Terara | 11.485349N, 37.102708E/0km | 1 | 1370 | 893.5 | 1.54 | 109.7 | <0.001 |
| Secondary cluster5 | Kalu | Jerjero (023) | 11.209294N, 39.904368E/0km | 1 | 995 | 726.6 | 1.37 | 44.6 | <0.001 |
| Secondary cluster6 | Bahir Dar Zuria | Yeginid | 11.443215N, 37.565428E/0km | 1 | 1036 | 866.9 | 1.20 | 15.6 | <0.001 |

*All kebeles of Metema and Gendawuha districts;

**Kurargenet, Addis amba Chelia

RR = Relative Risk; LLR = Log Likelihood Ratio

In Aneded and Awabel districts, the spatial cluster analysis identified one most likely type of cluster (LLR = 5588.9; p-value <0.001) with nine locations centered at Dimamelese kebele (10.049228N, 38.016623 E/ 10.9 km). five secondary significant clusters with five locations were identified in the two districts (Table 5).

## Purely temporal clusters of malaria cases

In the study areas, a significantly higher rate of purely temporal malaria cases was detected. The purely temporal cluster analysis of malaria cases detected three peak periods between September 2013 and November 2015 with LLR = 8727.5; p<0.00 (Fig 7).

## Spatiotemporal clusters of malaria cases

The most likely spatiotemporal cluster of malaria cases was detected in the Metema district at 22 locations with LLR = 97494.3, P-value <0.001 from June 2014 to November 2016. Secondary clusters of malaria cases were identified in all districts except in Tehulederie district with varying locations during September, October, November, and December in 2013 and 2014 (Table 6).

## Discussion

The result of this study showed a declining trend in malaria incidence both in the intervention and non-intervention sites during the study period. On average, a significant proportion (13.6%) of malaria incidence reduction was observed in the intervention sites as compared to the non-intervention. A statistically significant variation in malaria distribution was observed in space, time, and space-time at the intervention and non-intervention areas.

A blend of statistical methods, including scan statistical methods using ArcGIS and SaTS-can[TM] software, were used to examine the spatial, temporal, spatiotemporal, and hotspot clusters of malaria cases in 2012 kebeles under eight districts between epi week 37/2013 and 38/2018 (September 2013 through September 2018). In addition to the scan statistical methods, trend and seasonal decomposition analyses were performed using IBM SPSS and MS Excel software. We used the ARIMA model to evaluate the seasonal variation, seasonal irregularity, and trend component of time-series malaria data in the study period.

In line with other studies [52, 53], the trend and seasonal decomposition analyses of time-series data showed a decline in malaria incidence both in the intervention and non-intervention kebeles during the study period. With a unit increase in time (month), on average, the

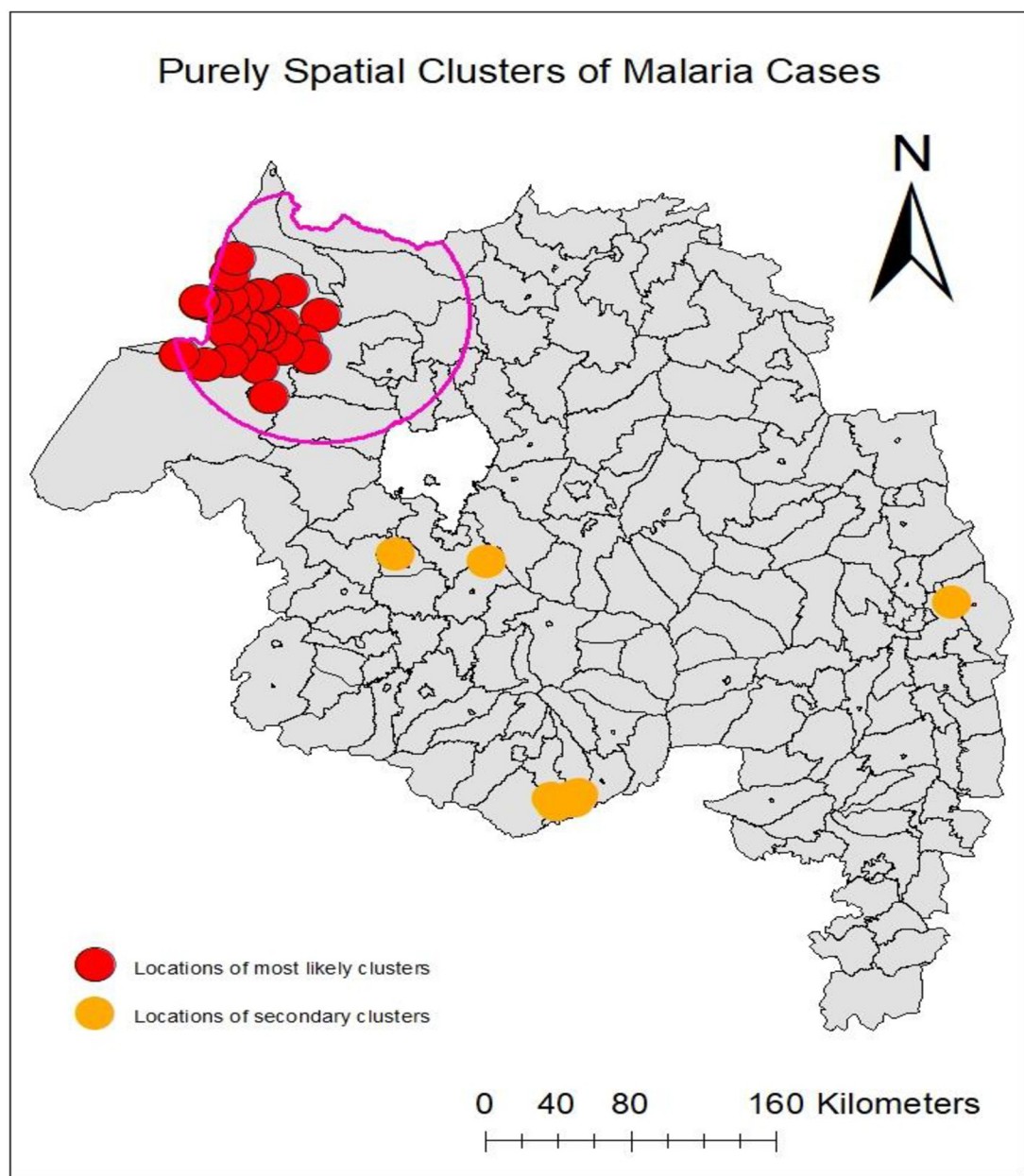

**Fig 6. Purely spatial clusters of malaria cases were detected using SaTScan[TM] in the Amhara Regional State, Ethiopia between 2013 and 2018. Copyright:** © 2022 Zeleke et al. The data on the map is derived from CSA, Ethiopia and APHI, Bahir Dar.

**Table 2. Most likely and secondary clusters of malaria cases in Metema and Gendawuha districts between epi week 37/2013 and 38/2018.**

| Cluster type | District | Kebele | Coordinates/Radius | Locations | Obs. cases | Exp. cases | RR | LLR | P-value |
|---|---|---|---|---|---|---|---|---|---|
| **Most likely cluster** | Metema | Meka | 12.673440N, 36.526005E/0km | 1 | 19699 | 5207.5 | 4.25 | 12541.5 | <0.001 |
| **Secondary cluster1** | Metema | Metemayohannes 01* | 12.941737N, 36.101129E/19.6km | 5 | 44856 | 24892.4 | 2.19 | 8335.5 | <0.001 |
| **Secondary cluster2** | Metema | Mesheha | 12.974605N, 36.411371E/0km | 1 | 14913 | 6117.1 | 2.61 | 4794.9 | <0.001 |

*Metemayohannes 02 & 03, Mender 6 7 8, Kokit Town

**Table 3. Most likely and secondary clusters of malaria cases in Bahir Dar Zuria and Mecha districts between epi week 37/2013 and 38/2018.**

| Cluster type | District | Kebele | Coordinates/Radius | Locations | Obs. cases | Exp. cases | RR | LLR | P-value |
|---|---|---|---|---|---|---|---|---|---|
| **Most likely cluster** | Bahir Dar Zuria | Yeginid* | 11.443215N, 37.565428E/10.3km | 4 | 2484 | 646.6 | 4.26 | 1599.1 | <0.001 |
| **Secondary cluster1** | Mecha | Tekle Terara | 11.485349N, 37.102708E/0km | 1 | 1370 | 224.7 | 6.49 | 1366.4 | <0.001 |
| **Secondary cluster2** | Bahir Dar Zuria | Debranta** | 11.768822N, 37.264675E/16.4km | 7 | 3210 | 1676.0 | 2.10 | 620.7 | <0.001 |
| **Secondary cluster3** | Mecha | Addis Lidet | 11.439752N, 37.064959E/0km | 1 | 486 | 120.5 | 4.11 | 315.9 | <0.001 |
| **Secondary cluster4** | Mecha | Birakat | 11.254615N, 37.174027E/0km | 1 | 518 | 173.6 | 3.04 | 225.1 | <0.001 |
| **Secondary cluster5** | Bahir Dar Zuria | Yinesa Sositu | 11.527069N, 37.310445E/0km | 1 | 630 | 284.0 | 2.26 | 159.1 | <0.001 |
| **Secondary cluster6** | Bahir Dar Zuria | Yemoshet/ Andassa | 11.552470N, 37.511745E/6.2km | 2 | 1021 | 563.7 | 1.86 | 154.8 | <0.001 |
| **Secondary cluster7** | Mecha | Dagi Abiyot | 11.309655N, 37.202332E/0km | 1 | 668 | 390.0 | 1.74 | 83.5 | <0.001 |
| **Secondary cluster8** | Mecha | Rim/Dil Betgil | 11.271923N, 37.288196E/4.9km | 2 | 824 | 561.5 | 1.49 | 55.4 | <0.001 |
| **Secondary cluster9** | Bahir Dar Zuria | Aluhayi | 11.381549N, 37.350409E/0km | 1 | 335 | 188.2 | 1.79 | 46.9 | <0.001 |
| **Secondary cluster10** | Bahir Dar Zuria | Maqual | 11.405233N, 37.442460E/0km | 1 | 359 | 213.1 | 1.70 | 41.9 | <0.001 |
| **Secondary cluster11** | Bahir Dar Zuria | Sebatamit | 11.534649N, 37.402749E/0km | 1 | 380 | 273.4 | 1.40 | 18.8 | <0.001 |
| **Secondary cluster12** | Mecha | Tatek Lesira | 11.344588 N, 37.079478 E/0 km | 1 | 329 | 248.6 | 1.33 | 12.0 | <0.001 |
| **Secondary cluster13** | Mecha | Anorayita | 11.273049 N, 37.009961 E/0 km | 1 | 260 | 192.6 | 1.35 | 10.7 | 0.001 |

*Wojir, Yemekat, Betemariam;

**Seqelet, Lijomie, Wonjeta, Lata Amba, Yigodi, Deq

incidence of malaria decreased by 0.044 in the intervention kebeles, whereas malaria incidence decreased by 0.038 in the non-intervention kebeles. This difference in the reduction of malaria incidence might be due to the effect of parasite clearance interventions in the elimination targeted areas. However, the difference was not statistically significant and warrants further evaluation of the effect of parasite clearance interventions on malaria incidence is essential to inform high-level decision-makers, program managers, and partners who are engaged in the malaria elimination program.

The global spatial autocorrelation of malaria incidence showed malaria transmission was not randomly distributed across the study areas and periods. The purely spatial cluster and hotspot/cold spot analyses in this study identified a statistically significant most likely type of clusters, secondary clusters, and hotspot clusters of malaria cases in the community. This finding agreed with the existing literature at the global, regional, and local scales [19, 21, 29, 31, 39, 54–56].

**Table 4. Most likely and secondary clusters of malaria cases in Kalu and Tehulederie districts between epi week 37/2013 and 38/2018.**

| Cluster type | District | Kebele | Coordinates/Radius | Locations | Obs. cases | Exp. cases | RR | LLR | P-value |
|---|---|---|---|---|---|---|---|---|---|
| **Most likely cluster** | Kalu | Jerjero (023) | 11.209294N, 39.904368E/0km | 1 | 995 | 108.8 | 10.61 | 1380.1 | <0.001 |
| **Secondary cluster1** | Kalu | Harbu 01 and 02 | 10.923421N, 39.785837E/1.9km | 2 | 1191 | 392.7 | 3.49 | 577.5 | <0.001 |
| **Secondary cluster2** | Kalu | Kurifa (035) | 10.910341N, 39.691119E/0km | 1 | 293 | 64.4 | 4.71 | 219.3 | <0.001 |
| **Secondary cluster3** | Kalu | Mudi Kalu (026) | 11.271761N, 39.845771E/0km | 1 | 349 | 97.1 | 3.74 | 199.6 | <0.001 |
| **Secondary cluster4** | Kalu | Resa (016) | 11.038924N, 39.924318E/0km | 1 | 333 | 152.3 | 2.25 | 82.4 | <0.001 |
| **Secondary cluster5** | Kalu | Keteteya (024) | 11.216153N, 39.858929E/0km | 1 | 340 | 162.3 | 2.15 | 76.2 | <0.001 |
| **Secondary cluster6** | Kalu | Arabo (021) | 11.153332N, 39.906370E/0km | 1 | 164 | 65.3 | 2.55 | 53.0 | <0.001 |
| **Secondary cluster7** | Kalu | Gerba 01/Wedajo (022) | 11.169611N, 39.936013 /0.2 km | 2 | 416 | 271.2 | 1.57 | 34.9 | <0.001 |
| **Secondary cluster8** | Kalu | Weraba tulu (032) | 10.928693N, 39.747937E/0 km | 1 | 192 | 114.3 | 1.70 | 22.4 | <0.001 |
| **Secondary cluster9** | Tehulederie | Muti Belig | 11.380169N, 39.729815 E/0 km | 1 | 165 | 94.8 | 1.76 | 21.6 | <0.001 |
| **Secondary cluster10** | Kalu | Agamsa (02) | 10.903907N, 39.849411E/0 km | 1 | 145 | 87.5 | 1.67 | 16.0 | <0.001 |
| **Secondary cluster11** | Tehulederie | Seglen | 11.253936N, 39.420264E/0 km | 1 | 169 | 107.4 | 1.59 | 15.3 | <0.001 |

**Table 5. Most likely and secondary clusters of malaria cases in Aneded and Awabel districts between epi week 37/2013 and 38/2018.**

| Cluster type | District | Kebele | Coordinates/Radius | Locations | Obs. cases | Exp. cases | RR | LLR | P-value |
|---|---|---|---|---|---|---|---|---|---|
| **Most likely cluster** | Awabel | Dimamelese* | 10.049228N, 38.016623E/10.9km | 9 | 6850 | 1817.4 | 7.49 | 5588.9 | <0.001 |
| **Secondary cluster1** | Aneded | Talaq Amba | 10.119253N, 37.915363E/0km | 1 | 540 | 241.5 | 2.30 | 139.9 | <0.001 |
| **Secondary cluster2** | Aneded | Tiquradebir | 10.272933N, 37.804131E/0km | 1 | 344 | 174.9 | 2.00 | 64.8 | <0.001 |
| **Secondary cluster3** | Aneded | Yewush | 10.167373N, 37.820535E/0km | 1 | 254 | 139.9 | 1.83 | 38.0 | <0.001 |
| **Secondary cluster4** | Aneded | Ayidbis Chendefo | 10.162268N, 37.942093E/0 km | 1 | 235 | 151.5 | 1.56 | 20.0 | <0.001 |
| **Secondary cluster5** | Awabel | Shebila Abeqestit | 10.299175N, 38.068320E/0 km | 1 | 289 | 198.7 | 1.47 | 18.3 | <0.001 |

*Kurargenet, Addis Amba, Dereqafer, Mizanwasha, Mekides, Tsidmariam, Amaya, Malgash

In this study, we considered the ecological-epidemiological transmission variations (high, moderate, low, and very low transmission settings) in the spatial scan statistical analysis. A separate scan statistical analysis of districts with moderate, low, and very low transmission settings identified an additional most-likely type of cluster, and secondary clusters of malaria cases. Whereas, in the high malaria transmission settings, the number of detected secondary clusters decreased with a location change of the most-likely type cluster. Thus, a separate scan statistical analysis needs to be considered when analyzing data collected from different transmission settings.

The purely spatial cluster analysis using weekly, monthly, and quarterly data has shown that there was no difference in detecting clusters of malaria cases in the study areas. This could be the spatial cluster analysis did not consider the time frame. Therefore, the time frame is not important while performing purely spatial cluster analysis [51, 57].

The purely temporal cluster analysis detected three peak periods identified in all study locations between September 2013 and November 2015. The peak malaria cases were observed in

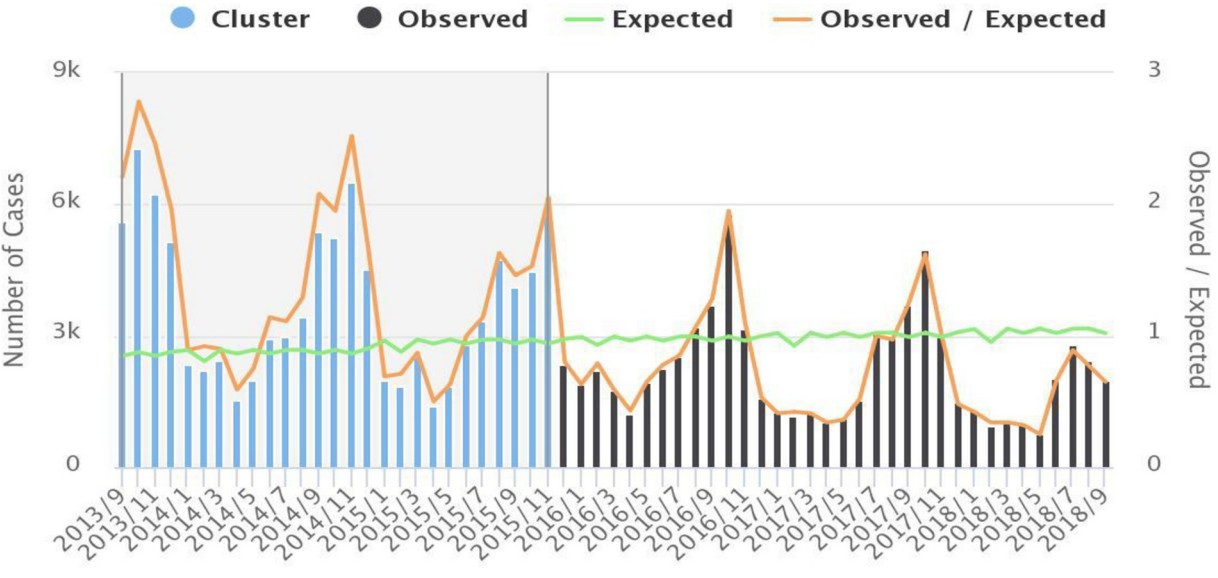

**Fig 7. Purely temporal clusters of malaria cases in the study areas between 2013/09 and 2018/09.**

**Table 6. Spatiotemporal clusters of malaria cases in the study areas, between 2013/09 to 2018/.**

| Cluster type | District | Kebele | Coordinates/Radius | Time frame | Obs. cases | Exp. cases | RR | LLR | p-value |
|---|---|---|---|---|---|---|---|---|---|
| **Most likely cluster** | Metema/ Gendawuha | Das Michael* | 12.762526N, 36.245767E/ 37.4 km | 2014/6/1 to 2016/ 11/30 | 72213 | 9498.2 | 12.23 | 97494.3 | <0.001 |
| **Secondary cluster1** | Awabel | Dimamelese/Kurargenet, Addis amba | 10.049228N, 38.016623E/ 4.9 km | 2014/11/1 to 2014/12/31 | 833 | 81.6 | 10.25 | 1185.4 | <0.001 |
| **Secondary cluster2** | Bahir Dar Zuria | Yeginid/Wojir/Yemekat/ Betemariam | 11.443215N, 37.565428E/ 10.3km | 2013/10/1 to 2013/12/31 | 1150 | 179.5 | 6.44 | 1168.1 | <0.001 |
| **Secondary cluster3** | Aneded | Shumburma, Malgash, Talaq Amba | 10.065235N, 37.897291E/ 6.3km | 2013/9/1 to 2013/ 12/31 | 956 | 198.3 | 4.84 | 747.9 | <0.001 |
| **Secondary cluster4** | Bahir Dar Zuria | Wonjeta | 11.683347N, 37.282703E/0 km | 2013/9/1 to 2014/ 4/30 | 671 | 251.9 | 2.67 | 238.7 | <0.001 |
| **Secondary cluster5** | Mecha | Birakat | 11.254615N, 37.174027E/0 km | 2013/9/1 to 2013/ 11/30 | 262 | 49.5 | 5.30 | 224.4 | <0.001 |
| **Secondary cluster6** | Kalu | Jerjero (023) | 11.209294 N, 39.904368 E/ 0 km | 2015/7/1 to 2015/ 11/30 | 282 | 64.7 | 4.36 | 198.0 | <0.001 |
| **Secondary cluster7** | Mecha | Tekle Terara | 11.485349N, 37.102708 E/ 0 km | 2014/9/1 to 2016/ 12/31 | 827 | 435.2 | 1.90 | 139.5 | <0.001 |
| **Secondary cluster8** | Mecha | Berhan Chora | 11.149402 N, 37.098154 E/ 0 km | 2014/10/1 to 2014/10/31 | 125 | 24.0 | 5.21 | 105.2 | <0.001 |
| **Secondary cluster9** | Mecha | Dagi Abiyot | 11.309655 N, 37.202332 E/ 0 km | 2013/9/1 to 2013/ 9/30 | 139 | 37.3 | 3.73 | 81.1 | <0.001 |
| **Secondary cluster10** | Mecha | Rim, Dil Betgil, Zemen Berhan | 11.271923 N, 37.288196 E/ 6.1 km | 2014/10/1 to 2014/11/30 | 269 | 118.7 | 2.27 | 69.9 | <0.001 |
| **Secondary cluster11** | Kalu | Harbu 01 | 10.923421 N, 39.785837 E/ 0 km | 2017/3/1 to 2017/ 5/31 | 167 | 72.1 | 2.32 | 45.4 | <0.001 |
| **Secondary cluster12** | Kalu | Mudi Kalu (026) | 11.271761 N, 39.845771 E/ 0 km | 2014/9/1 to 2015/ 3/31 | 191 | 88.5 | 2.16 | 44.5 | <0.001 |
| **Secondary cluster13** | Kalu | Resa (016) | 11.038924 N, 39.924318 E/ 0 km | 2013/10/1 to 2013/10/31 | 75 | 24.5 | 3.06 | 33.4 | <0.001 |
| **Secondary cluster14** | Bahir Dar Zuria | Yinesa Sositu | 11.527069 N, 37.310445 E/ 0 km | 2013/9/1 to 2013/ 11/30 | 156 | 79.4 | 1.97 | 28.8 | <0.001 |
| **Secondary cluster15** | Kalu | Kurifa (035) | 10.910341 N, 39.691119 E/ 0 km | 2014/9/1 to 2014/ 11/30 | 71 | 24.9 | 2.85 | 28.3 | <0.001 |
| **Secondary cluster16** | Mecha | Tatek Lesira | 11.344588 N, 37.079478 E/ 0 km | 2015/5/1 to 2015/ 5/31 | 57 | 19.4 | 2.93 | 23.8 | <0.001 |
| **Secondary cluster17** | Mecha | Addis Alem | 11.369247 N, 37.038942 E/ 0 km | 2014/9/1 to 2014/ 11/30 | 178 | 102.0 | 1.75 | 23.1 | <0.001 |
| **Secondary cluster18** | Bahir Dar Zuria | Aluhayi | 11.381549 N, 37.350409 E/ 0 km | 2013/9/1 to 2013/ 12/31 | 133 | 69.9 | 1.90 | 22.5 | <0.001 |
| **Secondary cluster19** | Mecha | Abiyot Fana | 11.212111 N, 37.083776 E/ 0 km | 2013/9/1 to 2013/ 11/30 | 163 | 96.3 | 1.69 | 19.1 | <0.001 |
| **Secondary cluster20** | Kalu | Keteteya (024) | 11.216153 N, 39.858929 E/ 0 km | 2013/10/1 to 2013/10/31 | 60 | 26.1 | 2.30 | 16.0 | 0.008 |

*Lemlem Terara, Agamwuha, Kokit Town, Kumer Aftit, Gendawuha 02, Gendawuha 01, Metemayohannes 03, Gubay Jejebit, Gendawuha Birshign, Mender 6 7 8, Diviko, Metemayohannes 02, Shinfa Town, Metemayohannes 01, Lencha, Zebach Bahir, Wodi Anbeso, Mesheha, Tumet Mendoka, Meka, Shemlegara

October and November which is supported by the seasonal decomposition of time-series data, and it occurred in the major malaria transmission season. The findings are in line with different studies conducted in Ethiopia [21, 39, 58]. This might be in the major malaria transmission season the climatic conditions are favorable for mosquitoes' breeding and life cycle of the malaria parasite in the mosquitoes.

The spatiotemporal cluster analysis identified a high variability of malaria transmission in space and time. The most likely type of spatiotemporal clusters were found in Gendawuha and Metema districts between June 2014 and November 2016. This could be low utilization of malaria vector control interventions, population mobility to these districts, and climatic variations. Many of the detected most likely and secondary clusters were observed between September and December.

In this study, a mix of methods and models were used to understand the trend and seasonal variation of malaria transmission in the study areas and period. The use of different spatial cluster analysis tools (SaTScanTM and ArcGIS) makes the evidence stronger than using a single tool. The national malaria elimination program aims to eliminate malaria by the end of 2030 [6]. Therefore, large-scale additional evidence is essential for appropriate targeting of malaria elimination interventions to better achievement of the elimination goal.

For easy retrieval of data and to improve the data quality, the *DHIS2* reporting platform was found very useful. In this analysis, we used the malaria data only generated by the public health facilities and the treatment-seeking tendency of the community could make underestimate the actual burden of the malaria cases.

## Conclusions

The trend in malaria incidence was declining both in the intervention and non-intervention areas during the study period. A significant proportion of malaria incidence reduction was observed in the intervention areas. The difference could be the effect of the parasite clearance interventions this warrants further evaluation of the effect of parasite clearance interventions on malaria incidence is important to inform policymakers, program managers, and partners who are working on a malaria elimination program.

Malaria distribution has shown heterogeneity in space, time, and space-time both in the intervention and non-intervention areas. There was a statistically significant spatial, temporal, and spatiotemporal distribution of malaria in the community. Spatiotemporal variation of malaria guided decision-makers and program managers on the selection of appropriate parasite clearance intervention and wise allocation of scarce resources. Conducting further studies is essential to identify factors associated with clusters of malaria for better-targeted interventions. Detecting and understanding clusters of malaria infection at the hamlet and individual level will be helpful for the effective and efficient use of resources.

## Supporting information

**S1 File. Seasonal decomposition of malaria incidence in the intervention kebeles.**
(XLS)

**S2 File. Seasonal decomposition of malaria incidence in the non-intervention kebeles.**
(XLS)

## Acknowledgments

We are very grateful to the Amhara Public Health Institute and PATH for their dedication and commitment to collecting and providing the weekly *DHIS2* malaria data. We also thank the Central Statistics Agency (CSA), Ethiopia for providing shapefiles for the study areas. We would like to thank Bahir Dar University for providing the opportunity to study. We have no words to express our gratitude to Belendia Abdissa for his unreserved IT support.

## Author Contributions

**Conceptualization:** Melkamu Tiruneh Zeleke, Muluken Azage Yenesew.

**Formal analysis:** Melkamu Tiruneh Zeleke.

**Methodology:** Melkamu Tiruneh Zeleke, Kassahun Alemu Gelaye, Muluken Azage Yenesew.

**Software:** Melkamu Tiruneh Zeleke.

**Writing – original draft:** Melkamu Tiruneh Zeleke.

**Writing – review & editing:** Kassahun Alemu Gelaye, Muluken Azage Yenesew.

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
