## [Decision Letter · Decision Letter 0]

29 Apr 2022

PONE-D-22-05813Spatiotemporal Variation of Malaria Incidence in Parasite Clearance Interventions and Non-Intervention areas in the Amhara Regional State, Ethiopia.PLOS ONE

Dear Dr. Zeleke,

Thank you for submitting your manuscript to PLOS ONE. After careful consideration, we feel that it has merit but does not fully meet PLOS ONE’s publication criteria as it currently stands. Therefore, we invite you to submit a revised version of the manuscript that addresses the points raised during the review process. The suggestion of the reviewers can be assessment by authors. Please submit your revised manuscript by Jun 13 2022 11:59PM.

Please include the following items when submitting your revised manuscript:A rebuttal letter that responds to each point raised by the academic editor and reviewer(s). You should upload this letter as a separate file labeled 'Response to Reviewers'.A marked-up copy of your manuscript that highlights changes made to the original version. You should upload this as a separate file labeled 'Revised Manuscript with Track Changes'.An unmarked version of your revised paper without tracked changes. You should upload this as a separate file labeled 'Manuscript'.

We look forward to receiving your revised manuscript.

Kind regards,

José Luiz Fernandes Vieira

Academic Editor

PLOS ONE

Journal Requirements:

4. We note that Figure 1, 5 and 6 in your submission contain map images which may be copyrighted. All PLOS content is published under the Creative Commons Attribution License (CC BY 4.0), which means that the manuscript, images, and Supporting Information files will be freely available online, and any third party is permitted to access, download, copy, distribute, and use these materials in any way, even commercially, with proper attribution. For these reasons, we cannot publish previously copyrighted maps or satellite images created using proprietary data, such as Google software (Google Maps, Street View, and Earth). For more information, see our copyright guidelines: http://journals.plos.org/plosone/s/licenses-and-copyright.

 a. You may seek permission from the original copyright holder of Figure 1, 5 and 6 to publish the content specifically under the CC BY 4.0 license. 

5. Please include your tables as part of your main manuscript and remove the individual files. Please note that supplementary tables (should remain/ be uploaded) as separate "supporting information" files.

Reviewers' comments:

Reviewer's Responses to Questions

**Comments to the Author**

1. Is the manuscript technically sound, and do the data support the conclusions?

Reviewer #1: Partly

Reviewer #2: Yes

2. Has the statistical analysis been performed appropriately and rigorously? 

Reviewer #1: Yes

Reviewer #2: Yes

3. Have the authors made all data underlying the findings in their manuscript fully available?

Reviewer #1: No

Reviewer #2: Yes

4. Is the manuscript presented in an intelligible fashion and written in standard English?

Reviewer #1: Yes

Reviewer #2: Yes

5. Review Comments to the Author

Reviewer #1: I suggest that the article be rewritten, adding explanations about the geographical organization of the country, its administrative division. This understanding is necessary to be able to assess the selection of study regions.

Futhermore it was not very clear what interventions had been carried out. In addition, to having selected a small number of kebele in relation to the total studied.

Reviewer #2: The article has relevance and originality for publication by the journal, but needs adjustments for a better presentation:

The title is relevant, concise and aligned with the general objective of the work.

The abstract demonstrates that the research is relevant and has been constructed logically.

Suitable keywords

Introduction is contextualized, updated and presented in an objective and coherent way, leading the reader to the objective of the study.

Methodology was appropriate and satisfactory for the purpose of the study.

The results are presented and discussed in depth, and it is possible to qualify the study with the addition of illustrative figures and tables, facilitating the reading and analysis of the data. Note regarding the positioning of Figure 2 and Figure 3 in the work, where Figure 3 is before Figure 2.

Discussion is coherent and sufficient for a good understanding of the study. Punctual suggestions should be made by the authors in order to better qualify the article, such as changing the positioning of lines 338, 339 and 340 for Conclusion.

The authors conclude based on the results, but similarly to the discussion, it is recommended to remove lines 348, 349 and 350 from the Conclusion and insert them in discussion.

Up-to-date and consistent bibliography. All authors cited in the text were described in the reference.

My opinion is PARTIALLY SATISFACTORY recommending this study for publication in PLOS ONE, as soon as the authors comply with the comments indicated in the text.

6. PLOS authors have the option to publish the peer review history of their article (what does this mean?). If published, this will include your full peer review and any attached files.

Reviewer #1: No

Reviewer #2: No

---

## [Author Response · Author response to Decision Letter 0]

1 Jul 2022

Author’s Response to Academic Editor’s and Reviewers’ Comments:

Manuscript No: PONE-D-22-05813

Title: Spatiotemporal variation of malaria incidence in parasite clearance interventions and non-intervention areas in the Amhara Regional State, Ethiopia.

We would like to thank the academic editor and the reviewers for the thorough review of our manuscript, encouraging words, helpful comments, and the opportunity to resubmit a revised copy of the manuscript. We have addressed all the comments as suggested by the academic editor and reviewers in our manuscript, as indicated below: 

Response to Academic Editor’s Comments:

Comment:

Response:

The manuscript is revised using PLOS ONE’s style template found in the above links to meet the journal requirements, now the manuscript meets PLOS ONE’s style requirements (Please see the marked-up and unmarked copies of the revised documents). 

Comment:

Response:

The study was conducted using retrospective data which were collected using the DHIS2 reporting platform. The data were aggregated (no individual data was collected) and no individual identifiers were attached to the data. Now, the ethical consideration section of the manuscript is updated based on your comments. 

Comments: 

Response:

Due to third-party restrictions, data are available from the Amhara Public Health Institute (APHI). Interested researchers may contact the institute to get permission to access the data. You can contact the institute through email: admin@aphi.gov.et or aphi172008@gmail.com; Office phone number: +251582263227. In this revised manuscript, a minimal dataset uploaded as a supporting file after getting the permission. The data availability section of the manuscript is now updated in response to the editor’s comments and mentioned in the cover letter. 

Comments: 

4. We note that Figure 1, 5 and 6 in your submission contain map images which may be copyrighted. All PLOS content is published under the Creative Commons Attribution License (CC BY 4.0), which means that the manuscript, images, and Supporting Information files will be freely available online, and any third party is permitted to access, download, copy, distribute, and use these materials in any way, even commercially, with proper attribution. For these reasons, we cannot publish previously copyrighted maps or satellite images created using proprietary data, such as Google software (Google Maps, Street View, and Earth). For more information, see our copyright guidelines: http://journals.plos.org/plosone/s/licenses-and-copyright.

 a. You may seek permission from the original copyright holder of Figure 1, 5 and 6 to publish the content specifically under the CC BY 4.0 license. 

Response:

The issue regarding the copyright of the maps is valid and very important. I’m aware of that before using it. The maps are the official maps of the country and its lower-level structure. I used the maps to show the study areas and the result of the study (this is the only reason). The maps are published by the author, and the shapefiles I used to prepare the maps were the property of the Ethiopia Central Statistical Agency (CSA) after getting permission to use it. We acknowledge the CSA of Ethiopia for providing the shapefiles. 

Comments:

5. Please include your tables as part of your main manuscript and remove the individual files. Please note that supplementary tables (should remain/ be uploaded) as separate "supporting information" files. 

Response:

All the individual tables are now included in the main manuscript in response to the editor’s comments. Now there are no individual files in the revised submission. In case you want to get access to the individual files, you can find them in the first submission. 

Reviewers' comments:

Reviewer's Responses to Questions

Comments to the Author

1. Is the manuscript technically sound, and do the data support the conclusions?

Reviewer #1: Partly

Reviewer #2: Yes

Response:

Now, the manuscript is revised in response to the comments, and the minimum dataset as supporting information is uploaded in this revised manuscript. 

2. Has the statistical analysis been performed appropriately and rigorously?

Reviewer #1: Yes

Reviewer #2: Yes

Response: It is ok,

3. Have the authors made all data underlying the findings in their manuscript fully available?

Reviewer #1: No

Reviewer #2: Yes

Response: Due to third-party restrictions, data are available from the Amhara Public Health Institute (APHI). Interested researchers may contact the institute to get permission to access the data. You can contact them through email: admin@aphi.gov.et or aphi172088@gmail.com; Office phone number: +251582263227. In this revised manuscript, a minimal dataset is uploaded as a supporting file after getting permission. 

4. Is the manuscript presented in an intelligible fashion and written in standard English?

Reviewer #1: Yes

Reviewer #2: Yes

Response: It is ok. 

5. Review Comments to the Author

Reviewer #1: I suggest that the article be rewritten, adding explanations about the geographical organization of the country, its administrative division. This understanding is necessary to be able to assess the selection of study regions.

Futhermore it was not very clear what interventions had been carried out. In addition, to having selected a small number of kebele in relation to the total studied.

Response: In collaboration with the government and partners, malaria parasite clearance interventions (mass testing and treatment followed by focal testing and treatment) were implemented in six kebeles (the lowest administrative unit in Ethiopia) of eight districts of Amhara Regional State, Ethiopia (mentioned in the introduction section). The study map showed the geographic organization of the country (Fig 1). The intervention kebeles were selected purposively (Scott et al. Malar J).

Comment: 

Reviewer #2: The article has relevance and originality for publication by the journal, but needs adjustments for a better presentation:

The title is relevant, concise and aligned with the general objective of the work.

The abstract demonstrates that the research is relevant and has been constructed logically.

Suitable keywords

Introduction is contextualized, updated and presented in an objective and coherent way, leading the reader to the objective of the study.

Methodology was appropriate and satisfactory for the purpose of the study.

The results are presented and discussed in depth, and it is possible to qualify the study with the addition of illustrative figures and tables, facilitating the reading and analysis of the data. Note regarding the positioning of Figure 2 and Figure 3 in the work, where Figure 3 is before Figure 2.

Discussion is coherent and sufficient for a good understanding of the study. Punctual suggestions should be made by the authors in order to better qualify the article, such as changing the positioning of lines 338, 339 and 340 for Conclusion.

The authors conclude based on the results, but similarly to the discussion, it is recommended to remove lines 348, 349 and 350 from the Conclusion and insert them in discussion.

Up-to-date and consistent bibliography. All authors cited in the text were described in the reference.

My opinion is PARTIALLY SATISFACTORY recommending this study for publication in PLOS ONE, as soon as the authors comply with the comments indicated in the text.

Response: Regarding the position of figures, (Fig 2) is about the seasonal decomposition of malaria incidence in the intervention kebeles whereas (Fig 3) is about the seasonal decomposition of malaria incidence in the non-intervention kebeles. We can change the position of the figures if it is logical. Lines 338, 339, and 340 of the discussion are removed and inserted in the conclusion section of the manuscript. Lines 348, 349, and 350 are rephrased and kept in the original place to conclude the spatiotemporal distribution of malaria. If we removed and inserted the paragraph in the discussion section, the manuscript lacks a conclusion regarding the spatiotemporal distribution of malaria. Now, comments are considered, and the manuscript is updated in response to the reviewers’ comments. 

Thank you all for your comments!

---

## [Decision Letter · Decision Letter 1]

30 Aug 2022

Spatiotemporal variation of malaria incidence in parasite clearance interventions and non-intervention areas in the Amhara Regional State, Ethiopia.

PONE-D-22-05813R1

Dear Dr. Zeleke,

We’re pleased to inform you that your manuscript has been judged scientifically suitable for publication and will be formally accepted for publication once it meets all outstanding technical requirements.

Kind regards,

José Luiz Fernandes Vieira

Academic Editor

PLOS ONE

Additional Editor Comments (optional):

Reviewers' comments:

Reviewer's Responses to Questions

**Comments to the Author**

1. If the authors have adequately addressed your comments raised in a previous round of review and you feel that this manuscript is now acceptable for publication, you may indicate that here to bypass the “Comments to the Author” section, enter your conflict of interest statement in the “Confidential to Editor” section, and submit your "Accept" recommendation.

Reviewer #1: All comments have been addressed

2. Is the manuscript technically sound, and do the data support the conclusions?

Reviewer #1: (No Response)

3. Has the statistical analysis been performed appropriately and rigorously? 

Reviewer #1: (No Response)

4. Have the authors made all data underlying the findings in their manuscript fully available?

Reviewer #1: (No Response)

5. Is the manuscript presented in an intelligible fashion and written in standard English?

Reviewer #1: (No Response)

6. Review Comments to the Author

Reviewer #1: (No Response)

7. PLOS authors have the option to publish the peer review history of their article (what does this mean?). If published, this will include your full peer review and any attached files.

Reviewer #1: No

---

## [Editor Report · Acceptance letter]

8 Sep 2022

PONE-D-22-05813R1 

Spatiotemporal variation of malaria incidence in parasite clearance interventions and non-intervention areas in the Amhara Regional State, Ethiopia.  

Dear Dr. Zeleke:

I'm pleased to inform you that your manuscript has been deemed suitable for publication in PLOS ONE. Congratulations! Your manuscript is now with our production department. 

Kind regards, 

on behalf of

Dr. José Luiz Fernandes Vieira 

Academic Editor

PLOS ONE